# Cryo-EM of full-length α-synuclein reveals fibril polymorphs with a common structural kernel

Binsen Li[1], Peng Ge[2], Kevin A. Murray[3], Phorum Sheth[1], Meng Zhang[3], Gayatri Nair[1], Michael R. Sawaya [3], Woo Shik Shin[1], David R. Boyer [3], Shulin Ye[2], David S. Eisenberg [3], Z. Hong Zhou[2,4] & Lin Jiang [1]

α-Synuclein (aSyn) fibrillar polymorphs have distinct in vitro and in vivo seeding activities, contributing differently to synucleinopathies. Despite numerous prior attempts, how polymorphic aSyn fibrils differ in atomic structure remains elusive. Here, we present fibril polymorphs from the full-length recombinant human aSyn and their seeding capacity and cytotoxicity in vitro. By cryo-electron microscopy helical reconstruction, we determine the structures of the two predominant species, a rod and a twister, both at 3.7 Å resolution. Our atomic models reveal that both polymorphs share a kernel structure of a bent β-arch, but differ in their inter-protofilament interfaces. Thus, different packing of the same kernel structure gives rise to distinct fibril polymorphs. Analyses of disease-related familial mutations suggest their potential contribution to the pathogenesis of synucleinopathies by altering population distribution of the fibril polymorphs. Drug design targeting amyloid fibrils in neurodegenerative diseases should consider the formation and distribution of concurrent fibril polymorphs.

[1] Department of Neurology, David Geffen School of Medicine, UCLA, Los Angeles, CA 90095, USA. [2] California Nano Systems Institute, UCLA, Los Angeles, CA 90095, USA. [3] Departments of Biological Chemistry and Chemistry and Biochemistry, Howard Hughes Medical Institute, UCLA-DOE Institute, UCLA, Los Angeles, CA 90095, USA. [4] Department of Microbiology, Immunology and Molecular Genetics, UCLA, Los Angeles, CA 90095, USA. These authors contributed equally: Binsen Li, Peng Ge, Kevin A. Murray. Correspondence and requests for materials should be addressed to D.S.E. (email: david@mbi.ucla.edu) or to Z.H.Z. (email: hong.zhou@ucla.edu) or to L.J. (email: jianglin@ucla.edu)

α-Synuclein (aSyn) is an intrinsically disordered protein, which can aggregate into different fibril forms, termed polymorphs. Polymorphic aSyn fibrils can recruit and convert native aSyn monomers into the fibril state, a process known as seeding[1]. Seeding of aSyn is associated with its pathological spread in the brain, contributing to multiple neurodegenerative diseases known as synucleinopathies, including Parkinson's disease (PD), dementia with Lewy bodies, and multiple system atrophy (MSA)[2,3].

Different aSyn fibril polymorphs have shown distinct seeding capacities in vitro and in vivo. Negative-stain electron microscopy (EM) images of aSyn fibrils extracted from PD and MSA patient brain tissues revealed fibril polymorphs with different widths: a major population of 10-nm-wide straight or twisted filaments and a minor population of 5-nm-wide straight filaments[2,3]. An additional EM study of recombinant aSyn fibrils confirmed the presence of similar fibril polymorphs, where each of the ~10-nm-wide filaments was composed of a bundle of two aSyn filaments[4]. More recently, two in vitro generated polymorphic fibrils (named ribbons and fibrils) exhibit different toxicity and in vitro[5] and in vivo[6] seeding properties. Peng et al.[7] demonstrated that brain-derived aSyn fibrils from different synucleinopathies are distinct in seeding potencies, which is consistent with the progression rate of each disease. In order to better understand the molecular basis for toxicity and seeding efficiency of aSyn aggregation in vitro and in vivo, atomic resolution structures of aSyn fibril polymorphs are crucially needed.

Previous studies have defined some structural details of aSyn fibrils. By micro-electron diffraction (microED)[8], structures of the preNAC region ($_{47}$GVVHGVTTVA$_{56}$) and NACore regions (non-amyloid-β component core, $_{68}$GAVVTGVTAVA$_{78}$), amyloidogenic segments critical for cytotoxicity and fibril formation, each revealed a pair of tightly mated in-register β-sheets forming a steric zipper. Moreover, a solid-state nuclear magnetic resonance (ssNMR) structure of recombinant aSyn revealed a Greek-key β-sheet motif in the hydrophobic core of a single fibril filament[9], where salt bridges (E46-K80), a glutamine ladder (Q79), and hydrophobic packing of aromatic residues (F94) contribute to the stability of the in-register β-sheet. These previous structural studies offer atomic insights into aSyn fibril architecture; however, additional structures are needed to elucidate the differences between aSyn fibril polymorphs. This information is necessary for the development of drugs targeting aSyn aggregation and seeding.

We set out to determine the structures of aSyn fibril species, and characterized one preparation of recombinant full-length aSyn containing various filamentous fibrils. The in vitro generated aSyn fibrils demonstrated a dose-dependent cytotoxicity and in vitro seeding in cells. Our cryo-EM study of the aSyn fibrils revealed two major polymorphs, termed rod and twister. Near-atomic structures (at a resolution of 3.7 Å) of both polymorphs showed a pair of β-sheet protofilaments sharing a conserved kernel consisting of a bent β-arch motif. However, the protofilaments of the structures contact with each other at different residue ranges, one at the NACore and the other at the preNAC region, forming different fibril cores. The involvement of NACore and preNAC steric zippers in the fibril cores of aSyn fibrils is supported by X-ray fiber diffraction experiments. In the rod and twister polymorphs, interface packing differences between the protofilaments lead to different fibril morphologies with distinct helical twists along the fibril axis. Structural analysis of disease-related mutations in the rod and twister structures suggests that aSyn fibril polymorphs may play different roles in aSyn aggregation and seeding.

## Results

**Seeding capacity and cytotoxicity of full-length human aSyn fibrils.** In order to produce a wide range of aSyn fibril polymorphs, we screened fibril growth conditions of full-length recombinant human aSyn (1–140) by varying pH, salt, and additives. All samples were incubated in quiescent conditions for 14–30 days, in order to best mimic the physiological conditions of in vivo fibril growth. Fibril growth was monitored using thioflavin T (ThT) aggregation kinetics. We confirmed the presence of a wide range of fibril morphologies using negative-stain EM (see Methods and Supplementary Fig. 1). One fibril preparation stood out with well-separated single filaments with or without an apparent twist (Fig. 1a and Supplementary Fig. 2) in the presence of tetrabutylphosphonium bromide (an ionic liquid additive used in protein crystallization) at room temperature. Two major populations in this fibril preparation, the straight and twisted filaments, were around 10 nm wide (Fig. 1b), which is consistent with the previously reported aSyn fibrils either generated in vitro or extracted from patient brains[4,10].

We performed biological experiments to assess the pathological relevance of the aSyn fibrils preparation. In vitro seeding of the fibrils was monitored using a biosensor cell assay. Human embryonc kidney 293T (HEK293T) cells endogenously express disease-associated aSyn A53T mutant fused with cyan fluorescent protein (CFP) or yellow fluorescent protein (YFP) as a fluorescence resonance energy transfer (FRET) pair[11]. The aSyn fibril seeds were transduced into cells and induced intracellular aSyn aggregation or inclusions that was quantified by flow cytometry-based FRET analysis[11] (see Methods and Supplementary Fig. 3). At a concentration of aSyn fibril seeds as low as 10 nM, we observed aSyn inclusions as fluorescent puncta in cells (white arrows in Fig. 1c). The quantified FRET signal indicated the level of cellular aggregation seeded by the aSyn fibrils followed a dose-dependent manner (Fig. 1d). We also characterized the cytotoxicity of these aSyn fibrils in differentiated PC12 cells. The aSyn fibrils used in in vitro seeding experiment showed a significant cytotoxicity at 500 nM (Fig. 1e). Thus, the aSyn fibrils used in this study were able to act as seeds and trigger intracellular amyloid aggregation and subsequent cytotoxicity.

**Cryo-EM structures of two aSyn fibril polymorphs.** We performed cryo-EM studies to further elucidate the structures of fibril polymorphs. Two-dimensional (2D) classification of the cryo-EM images revealed that the fibril preparation consisted of two major populations, as well as several minor ones (Supplementary Fig. 4). The two major populations were composed of two fibril polymorphs, herein referred to as "twister," which has a twist in its projection views, and "rod," which lacks an apparent twist. We determined the three-dimensional (3D) structures for both polymorphs to a resolution of 3.7 Å (Table 1, Fig. 2, and Supplementary Figs. 5, 6). Both structures consisted of two intertwined protofilaments related by an approximate $2_1$ screw axis of symmetry with a helical rise of 2.4 Å, which is consistent with the 2.4 Å reflection observed in the fiber diffraction patterns (Fig. 3e). The rod polymorph has a pitch of 920 Å and a right-handed helical twist of 179.5°; the twister polymorph has a shorter (460 Å) pitch and a right-handed helical twist of 179.1° (Fig. 2a).

We were able to build atomic models for both the rod and twister polymorphs, guided by side chain densities revealing distinct landmarks (Supplementary Figs. 7, 8). Both structures are composed of two protofilaments, each consisting of predominantly β-sheets. Out of the 140 amino acids in aSyn, 60 residues (L38-K97) are sufficiently ordered to be visible in the rod polymorph. As shown in the left panel of Fig. 2a, the polypeptide

chains stack into a Greek-key-like core with two turns, similar to the previous ssNMR-derived protofilament structure[9]. At both ends of the chain are lower-resolution densities that cannot be reliably modeled. In contrast, only 41 amino acids (K43-E83) are ordered in the twister polymorph, forming a bent β-arch (Fig. 2a, right panel). The more disordered chains at both termini project radially outward; they may account for the larger maximal width of the twister polymorph, as the ordered regions in both polymorphs have similar diameters.

**Unique inter-protofilament interfaces of the two polymorphs.** Comparison of the cryo-EM structures of the rod and twister polymorphs demonstrated the presence of a common protofilament kernel (root-mean-square deviation (RMSD) of residues H50-V77 = 2.2 Å for only Cα atoms, 2.5 Å for all atoms) (Fig. 3c, d). The twister polymorph has a well-ordered bent β-arch motif,

while the rod polymorph also has a bent β-arch but uses additional ordered residues to form a Greek-key-like fold. A large fraction of branched amino acid residues (Thr, Val) is involved in the mainly hydrophobic core of the bent β-arch (Fig. 3a, b). Major turns or bends in the backbones of the two structures coincide with the presence of glycines (G67, G84), stabilizing hydrogen bonds (N65 and G68, Q79 and G86), and solvent exposed charged residues (E57, K58) (Fig. 3a, b). A hydrophilic channel, lined by residues T54, T59, E61, T72, and T75 (Supplementary Fig. 9), is adjacent to the hydrophobic core in the center of both structures. The bent β-arch conformation represents a common protofilament kernel between the rod and twister polymorphs. Interestingly, the single protofilament structure in the ssNMR study[9] shows some similarities to the common protofilament kernel in cryo-EM structures, with an Cα RMSD of 3.4 Å (rod)

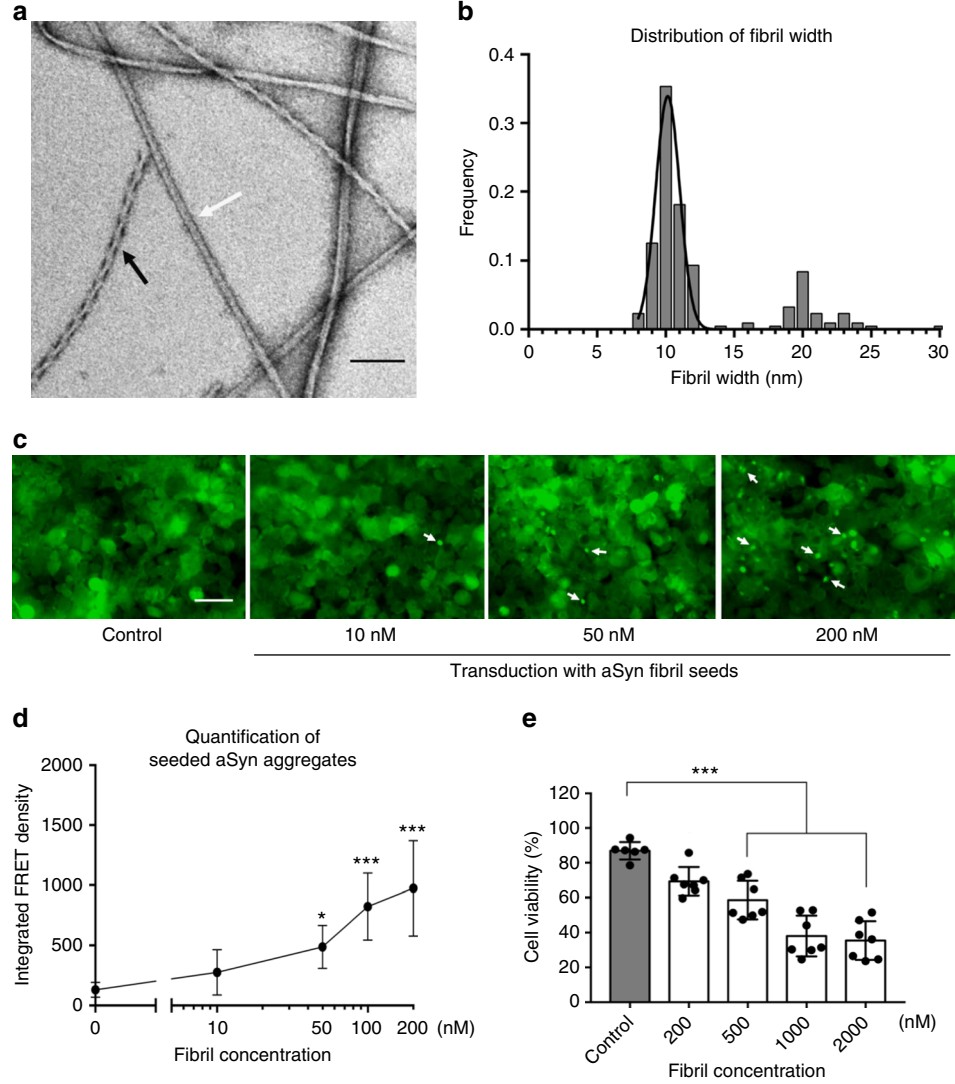

**Fig. 1** aSyn fibrils with distinct polymorphs have in vitro seeding and toxicity in cells. **a**, **b** Negative-stain EM (**a**) of full-length aSyn fibrils showing two distinct polymorphs—rod (non-twisted filaments, white arrow) and twister (twisted filaments, black arrow)—and a fibril width around 10 nm (**b**). **c**, **d** Direct visualization (**c**) and FRET-based quantification (**d**) of seeded intracellular aSyn aggregates. Fluorescent images obtained using the FITC channel (ex. 488 nm, em. 525 nm) showed aSyn aggregates as indicated by bright fluorescent puncta (white arrows in **c**). The diffuse background fluorescence came from endogenously expressed soluble, non-aggregated YFP-aSyn in the cells. Transduction of sonicated aSyn fibril seeds into the cells induced intracellular aSyn aggregation, which was also quantified using FRET analysis (**d** and Supplementary Fig. 3). **e** Cytotoxicity of aSyn fibrils evaluated by MTT-based cell viability assay of differentiated PC12, neuron-like cells. The aSyn fibrils used in the seeding experiment (**c**, **d**) have significant cytotoxicity ($p < 0.0001$) at 500 nM. Data are presented as mean ± standard error. Results are from multiple independent biological experiments with $n = 3$–5 per experiment. *$p \leq 0.01$, ***$p \leq 0.0001$ vs. control (buffer used to produce the aSyn fibrils). Scale bar: (a) 100 nm (c) 50 μm

**Table 1 Cryo-EM data collection, refinement, and validation statistics**

| | Rod polymorph EMD-7618 PDB:6CU7 | Twister polymorph EMD-7619 PDB:6CU8 |
|---|---|---|
| **Data collection** | | |
| Magnification | ×130,000 | ×130,000 |
| Defocus range (μm) | 1.5–4 | 1.5–4 |
| Voltage (kV) | 300 | 300 |
| Microscope | Titan Krios | Titan Krios |
| Camera | Gatan K2 Summit (GIF) | Gatan K2 Summit (GIF) |
| Frame exposure time (s) | 0.2 | 0.2 |
| No. of movie frames | 50 | 50 |
| Total electron dose (e⁻/Å⁻²) | 80 | 80 |
| Pixel size (Å) | 1.07 | 1.07 |
| **Reconstruction** | | |
| Box size (pixel) | 432 | 432 |
| Inter-box distance (Å) | 36 | 36 |
| No. of segments extracted | 182,253 | 182,253 |
| No. of segments after Class2D | 23,830 | 61,698 |
| No. of segments after Class3D | N/Aᵃ | 34,091 |
| Resolution | 3.7 | 3.7 |
| Map sharpening B-factor (Å²) | 100 | 100 |
| Helical rise (Å) | 2.40 | 2.40 |
| Helical twist (°) | 179.53 | 179.06 |
| **Atomic model** | | |
| No. of protein residues | 60 | 41 |
| Ramachandran plot values | | |
| Most favored (%) | 87.9 | 97.4 |
| Allowed (%) | 10.3 | 2.6 |
| Disallowed (%) | 1.72 | 0.0 |
| Rotamer outliers | 0.0 | 0.0 |
| RMS deviations | | |
| Bond lengths (Å) | 0.01 | 0.01 |
| Bond angles (°) | 0.89 | 0.87 |
| Clashscore | 25.67 | 22.39 |
| Map CC (whole unit cell) | 0.356 | 0.375 |
| Map CC (around atoms) | 0.754 | 0.727 |

N/A not applicable
ᵃParticles from Class2D are used in the refinement

the twister structure (Fig. 3g and Supplementary Fig. 11). We find generally good agreement between our energy calculations and the hypothesized effects each familial mutation may have on each polymorphic structure (Supplementary Fig. 12). An exception to this agreement is the H50Q mutation, where the prediction method fails to capture the complex H-bond networks of multiple residues H50, E47, and K45.

**Relevance of full-length aSyn fibrils with peptide zippers.** X-ray fiber powder diffraction of the full-length aSyn fibril polymorphs revealed cross-β fibril structures consistent with those of NACore and preNAC peptide fibrils (Fig. 3e). All fibril diffraction patterns contain a strong 4.7 Å reflection, characteristic of the stacking of β-strands along the fiber axis, and reflections near 8.0 and 11.5 Å, likely stemming from the staggering between adjacent β-sheets in the structure, either within a protofilament or between two protofilaments. All fibrils also have the reflection at 2.4 Å in their diffraction patterns. Observed in both cryo-EM structures, a helical rise of 2.4 Å, half the 4.8 Å spacing between β-strands, permits the two sheets to interdigitate tightly together. Similar 2.4 Å helical rises are observed in the microED structures of preNAC and NACore peptide fibrils[8]. This 2.4 Å reflection confirmed that the structures of aSyn fibrils and the peptide fibrils are all defined by an approximate 2₁ screw axis of symmetry. The resemblance of all of these fibril diffraction patterns suggested that the aSyn fibrils may share a fibril core in which NACore and preNAC are involved.

In the aSyn fibril preparation, the protofilaments in both the rod and twister structures share a conserved fibril kernel and contact with adjacent protofilaments at either preNAC or NACore regions. The rod polymorph has a longer pitch, while the twister polymorph has a pitch shorter by half. Distinct fibril morphologies indicated by fibril pitch thus arise from differences in packing, which are revealed in our near-atomic structures. Structural analysis of familial mutations in the rod and twister structures suggests that aSyn fibril polymorphs may play different roles in aSyn fibril formation in synucleinopathies.

**Discussion**
Protofilaments in aSyn fibrils are composed of single chains arranged in parallel in-register β-sheets. Fibril protofilaments can assemble in different arrangements to form several possible polymorphic structures. α-Synuclein fibrils isolated from PD patient brains have been shown to have polymorphic structures, with fibril widths of ~5 and ~10 nm[10]. Our cryo-EM structures of two polymorphs, each with a pair of protofilaments, are ~10 nm in width (99 Å for the rod structure and 96 Å for the twister structures, Fig. 2a). The single protofilament structure revealed in the ssNMR study was ~5 nm in width[9] and resembles the common protofilament kernel in cryo-EM structures of both rod and twister polymorphs, with an RMSD of 3.5 and 3.8 Å, respectively for the 38 matched residues (Supplementary Fig. 9). The recently published cryo-EM structure of a truncated aSyn (residues 1–121) fibril[19] has a structure similar to the rod polymorph of the full-length protein reported here (with an RMSD of 2.1 Å). Thus, different aSyn fibril polymorphs could arise from alternative arrangements of the same protofilament kernel. Similar phenomena have been observed in other amyloid proteins, including tau and β-amyloid, where different packing arrangements of the same protofilament kernel lead to polymorphic structures[20,21]. These observations suggest a generic mode of fibril architecture by the concurrent assembly of identical protofilaments (Fig. 4b).

Our structural studies reveal that the rod and the twister protofilaments assemble symmetrically about a homo-zipper of the preNAC segment or of the NACore segment, respectively.

and 3.4 Å (twister), respectively, for the 28 matched residues (Supplementary Fig. 10).

While a pair of identical protofilaments is intertwined in both structures, different steric zipper interfaces are present between the protofilaments. The highly complementary inter-protofilament interface in the rod polymorph, with a calculated shape complementary score[12] of 0.77, consists of a steric homo-zipper of the preNAC ($_{47}$GVVHGVTTVA$_{56}$) (Fig. 3c). The preNAC steric zipper in the rod structure is associated with six PD familial mutation sites (E46K, H50Q, G51D, A53E, A53T, and A53V; Fig. 4a)[13–18], with the potential to disrupt the preNAC zipper of fibril core in the rod structure (Fig. 3f). Based on the structural analysis (Supplementary Figs. 11 and 12), the mutation H50Q would interfere with the potential salt bridge E57-H50. The negative charge in the mutation G51D and A53E would likely disrupt the steric zipper interaction between the two protofilaments, while A53T and A53V would weaken the hydrophobic packing of the zipper.

In the twister structure, the interface between the two protofilaments (SC = 0.71) is a steric homo-zipper of the NACore ($_{68}$GAVVTGVTAVA$_{78}$) (Fig. 4c). The β-strands of the NACore interdigitate with each other and form the hydrophobic core, consisting of small apolar residues (A69, V71, V74). In the structure, the preNAC residues are located at the peripheral region away from the fibril core. Therefore, the six familial mutations of the preNAC region which potentially disrupt the rod structure may have little effect on the stability of

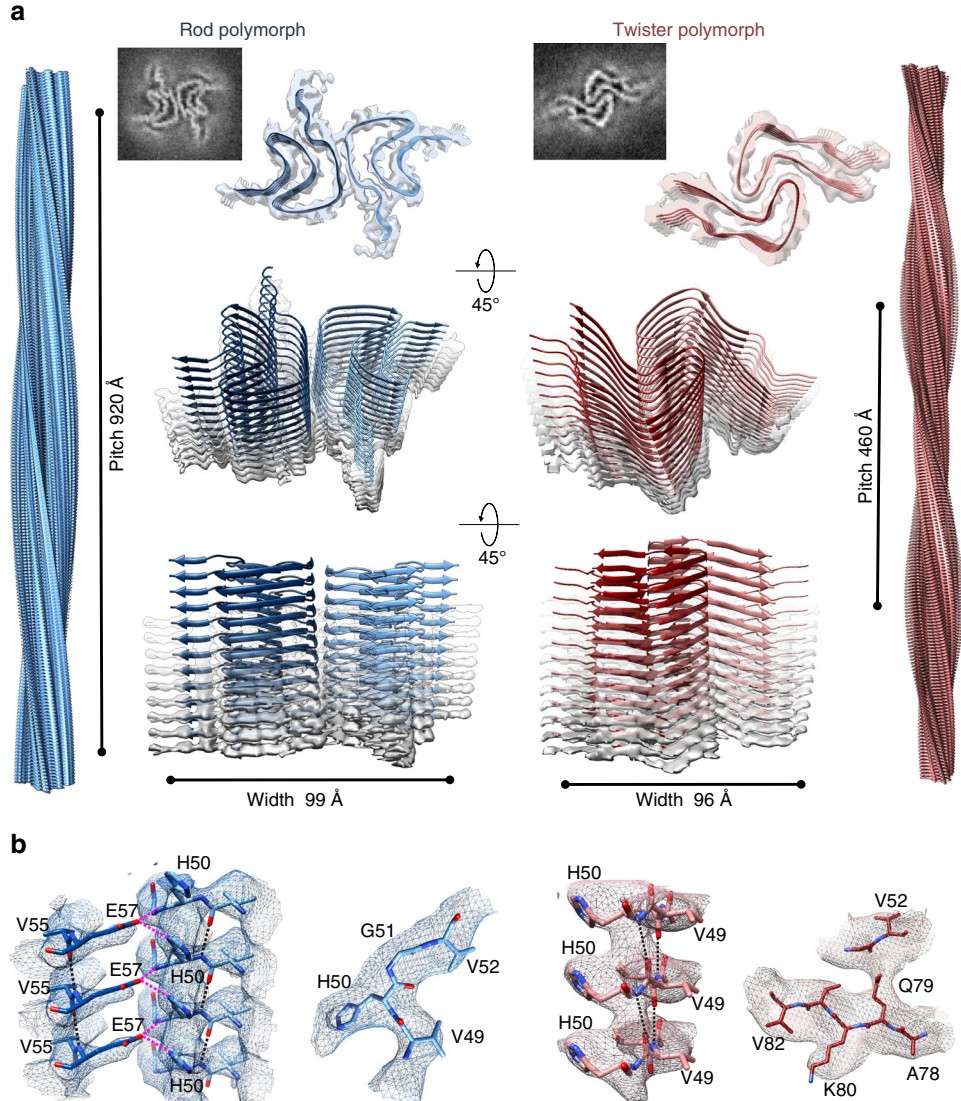

**Fig. 2** Cryo-EM structures and atomic models of the aSyn rod and twister polymorphs. **a** The cryo-EM structures of the rod (left) and twister (right) polymorphs of the full-length aSyn fibrils shown as density slices (top inlet), as semitransparent surfaces overlaid with their atomic models viewed from two different angles (lower panels). The rod (blue) and twister (red) polymorphs contain two protofilaments composed of stacked β-sheets and packed by an approximate $2_1$ screw axis of symmetry. Shown on the left and right sides are the 3D model of the rod and twister fibril polymorphs, respectively, with their distinctively different helical pitches depicted. **b** Model validation. Representative regions of density maps of both polymorphs are superimposed with their models showing match of side chain with cryo-EM densities. Intra-protofilament hydrogen bonds are shown in black dashed lines, and inter-protofilament hydrogen bonds are shown in magenta dashed lines. See details in Supplementary Figures 7 and 8

Since the twister and rod fibrils have structurally conserved kernels but contact at either the preNAC or NACore segments, the fibril polymorph is determined by the location of the protofilament packing interface instead of the kernel structures. The two structures of aSyn polymorphs revealed the steric homo-zipper core of the preNAC and NACore between the protofilaments. Together with the crucial contribution of the preNAC and NACore segments to the formation of aSyn fibrils, our structures present these unique protofilament interfaces as therapeutic targets to halt the fibrillization of aSyn. Atomic details of the preNAC and NACore zippers from our aSyn polymorphic structures provide insights in the structural-based designs of aSyn aggregation inhibitors in synucleinopathies.

Different aSyn fibril preparations, whether obtained from brain tissue or produced in vitro, may have different compositions of polymorphs. Each polymorph is distinguished by packing differences between protofilament kernels and makes distinct contributions to the biological activities of seeding and toxicity. The aSyn fibril preparation containing fibril polymorphs with different compositions thus could have discrete seeding efficiency and cytotoxicity profiles. Therefore, it is essential to characterize the biological function of each individual polymorph in order to understand the pathological role of the complex polymorphic fibrils.

The cryo-EM structure of the rod polymorph constructed around the fibril core of the preNAC region, and five PD familial mutations (E46K, H50Q, G51D, A53E, A53T, and A53V) are located at and associated to the preNAC region (Fig. 4a). Our structural analysis suggests that all these mutations would disfavor the fibril core of the rod structure without affecting the twister structure constructed around a different fibril core. Therefore, those point mutations would result in a different composition of polymorphic aSyn fibrils, by decreasing or eliminating the population of the rod polymorph while potentially

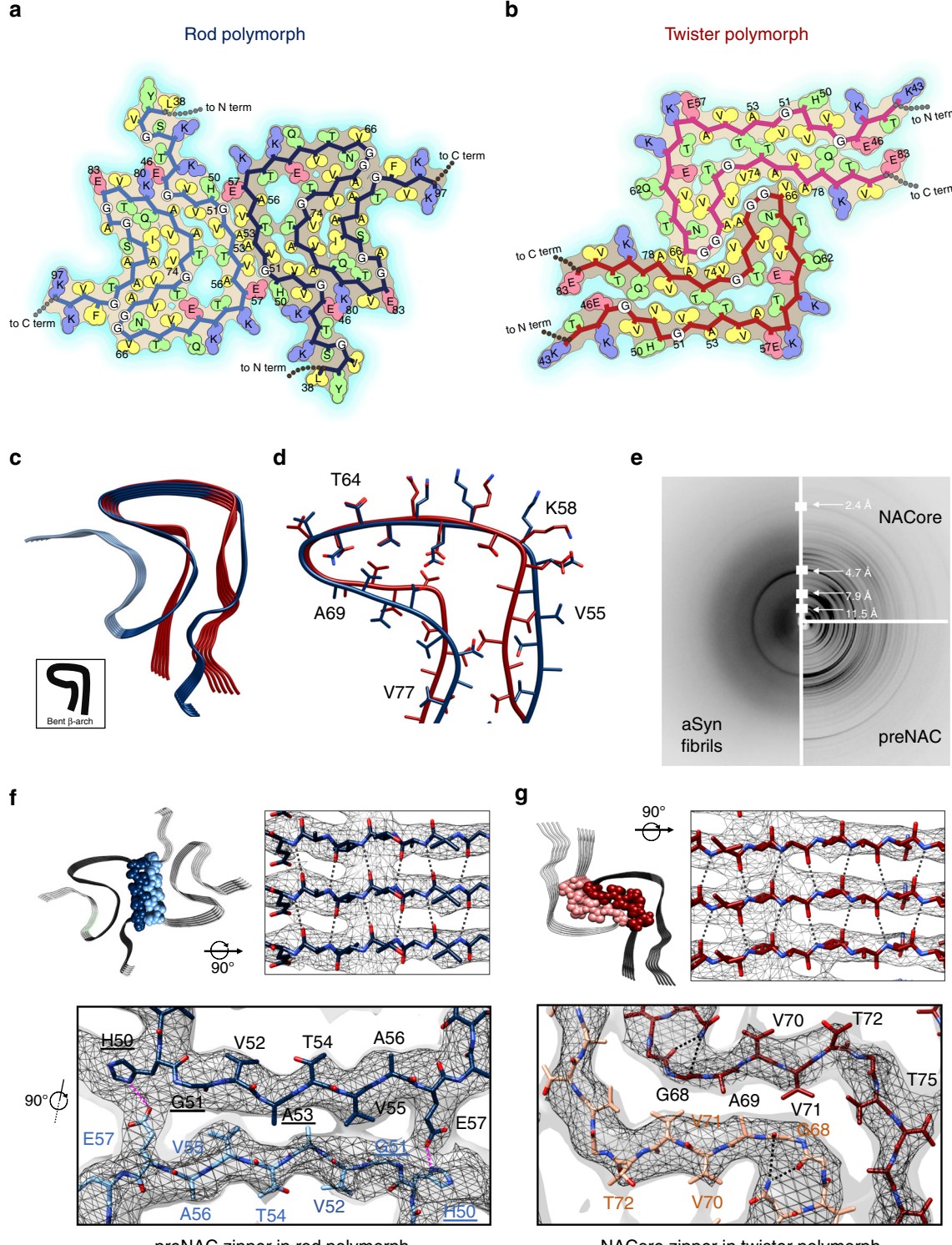

**Fig. 3** Distinct zipper interfaces between protofilament kernels in the two aSyn polymorphs. **a**, **b** Residue interactions of two asymmetric units in two opposing protofilaments elucidate packing between and within these two protofilaments in the rod (**a**) and twister (**b**) polymorphs (viewed down fibril axis). Residues are colored by hydrophobicity (yellow: hydrophobic; green: polar; red: negative charge; blue: positive charge). **c**, **d** An overlay of protofilaments of the rod (blue) and twister (red) polymorphs reveals a conserved kernel of a bent β-arch. **e** Diffraction patterns of the full-length aSyn fibrils agree with those of NACore and preNAC peptide fibrils. **f**, **g** The two protofilaments in the rod (**f**) and twister (**g**) polymorphs contact by different residues (space-filled) and have distinct fibril core of tightly packed steric zippers of preNAC (blue) and NACore (red), as previously observed in those peptide fibril structures. PD familial mutation residues are labeled with underlines. The cryo-EM density maps are shown as gray mesh surfaces. Intra-protofilament hydrogen bonds are shown in black dashed lines, and inter-protofilament hydrogen bonds are in magenta

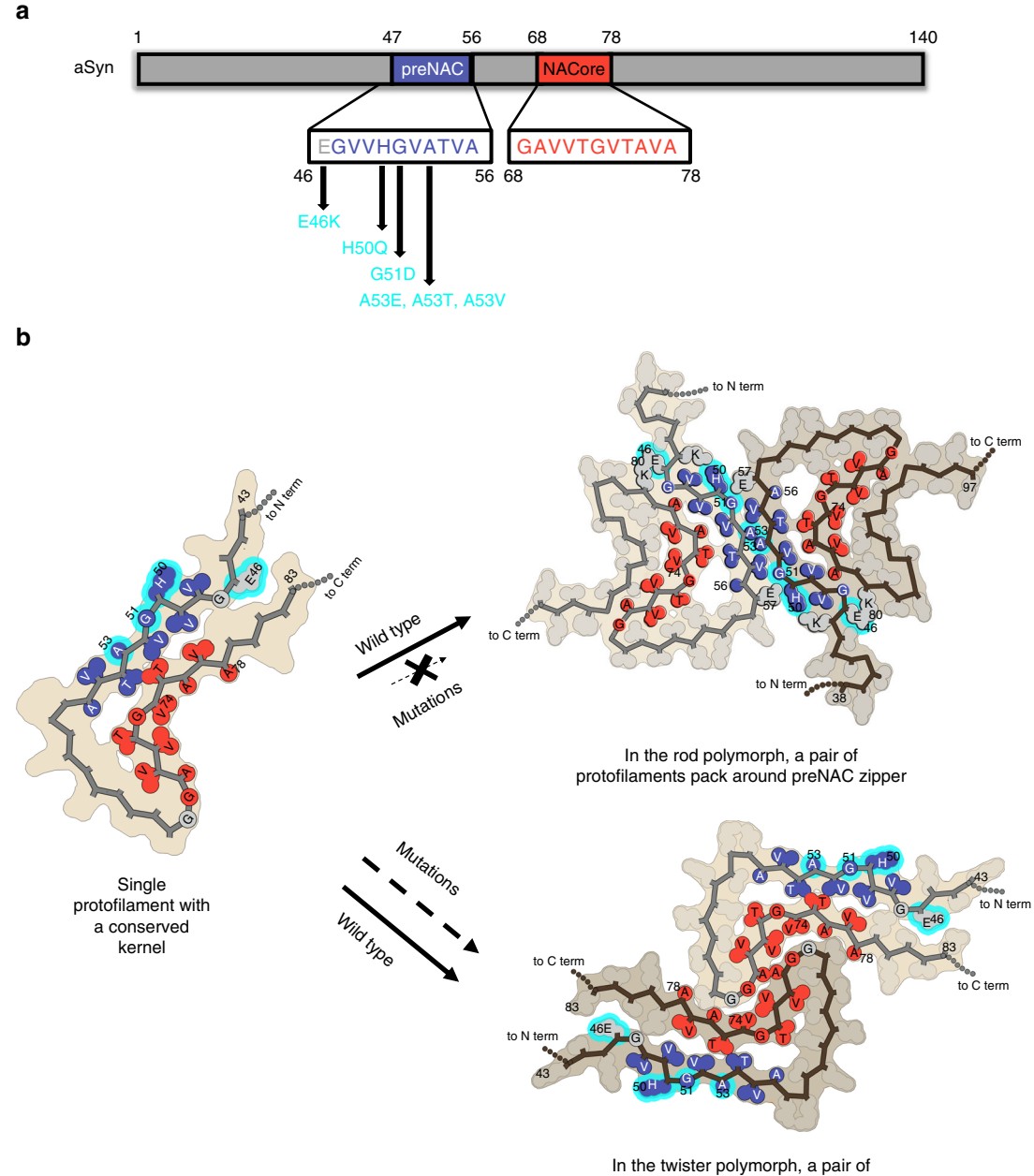

**Fig. 4** Morphogenesis of aSyn fibril polymorphs arising from inter-protofilament packing. **a** Primary sequence of preNAC (blue) and NACore (red) critical for the aggregation of aSyn 1–140 and six PD familial mutations (cyan) located near the preNAC region. **b** Protofilaments sharing a kernel structure of a bent β-arch assemble into the rod and twister fibril polymorphs by packing at preNAC and NACore zipper interfaces, respectively. The PD familial mutations (cyan) likely disfavor the rod structure over the twister structures, and alter the polymorphic composition of aSyn fibrils

inducing the formation of another fibril polymorph (Fig. 4b). The resulting changes in the ensembles of fibril polymorphs may alter their biological activity and underlie the phenotypic differences in patients with PD due to familial point mutations, suggesting aSyn fibril polymorphs have pathogenic contributions to synucleinopathies.

In summary, we have determined the cryo-EM structures of two fibril polymorphs of full-length recombinant aSyn with distinct protofilament interfaces. The rod and twister polymorphs are composed of protofilaments with highly conserved kernel structure assembled around different steric zipper interfaces, giving rise to polymorphism in aSyn fibrils. The two structures of fibril polymorphs elucidate atomic interactions of the steric zippers within the fibril cores, potentially guiding the future

drug design of aSyn aggregation inhibitors. These structural and functional studies thus establish the need to consider the contributions of all polymorphs and their relevance to overall pathogenesis when performing future rational design of therapeutic agents based on fibril structures.

## Methods

**Expression and purification of recombinant aSyn (1–140)**. Full-length aSyn protein was expressed in *Escherichia coli* (BL21-DE3 Gold strain, Agilent Technologies, Santa Clara, CA, USA) and purified according to a published protocol[8]. The bacterial induction started at an $OD_{600}$ of ~0.6 with 1 mM isopropyl β-D-1-thiogalactopyranoside for 6 h at 30 °C. The harvested bacteria were lysed with a probe sonicator for 10 min in an iced water bath. After centrifugation, the soluble fraction was heated in boiling water for 10 min and then titrated with HCl to pH 4.5 to remove the unwanted precipitants. After adjusting to neutral pH, the protein

was dialyzed overnight against Q Column loading buffer (20 mM Tris-HCl, pH 8.0). The next day, the protein was loaded onto a HiPrep Q 16/10 column and eluted using elution buffer (20 mM Tris-HCl, 1 M NaCl, pH 8.0). The eluent was concentrated using Amicon Ultra-15 centrifugal filters (Millipore Sigma) to ~5 mL. The concentrated sample was further purified with size-exclusion chromatography through a HiPrep Sephacryl S-75 HR column in 20 mM Tris, pH 8.0. The purified protein was dialyzed against water, concentrated to 3 mg/mL, and stored at 4°C. The concentration of the protein was determined using the Pierce™ BCA Protein Assay Kit (cat. no. 23225, Thermo Fisher Scientific).

**Fibril preparation monitored using ThT assay**. The fibril growth conditions were screened in the 96-well plate format in various pH, salts, and additives. Specifically, purified aSyn (100, 200, or 300 μM) was diluted in phosphate-buffered saline (PBS), 50 mM Tris buffer, or 5 mM Tris buffer at various pH (5.5, 6.5, 7.5, or 8.5) in the presence or absence of 24 commercially available crystal screening additives in the Ionic Liquid Screen (Hampton Research, Aliso Viejo, CA, USA). The samples were adequately mixed with 20 μM ThT and added into each well. The 96-well plates were incubated at either room temperature or 37 °C for 14–30 days. The ThT signal was monitored using the FLUOstar Omega Microplate Reader (BMG Labtech, Cary, NC, USA) at an excitation wavelength of 440 nm and an emission wavelength of 490 nm. Selected fibrils conditions from the ThT assay were used to grow the fibrils in the absence of ThT to be further characterized in the negative-stain EM. Out of hundreds of fibril growth conditions screened, one fibril growth condition (300 μM aSyn, 15 mM tetrabutylphosphonium bromide, room temperature) was selected for the rest of our study.

**Transmission electron microscopy**. The fibril sample (3 μL) was spotted onto a freshly glow-discharged carbon-coated electron microscopy grid. After 1 min, 6 μL uranyl acetate (2% in aqueous solution) was applied to the grid for 2 min. The excessive stain was removed by a filter paper. The samples were imaged using an FEI T12 electron microscope.

**Cryo-EM reconstruction and atomic modeling**. A 2.5-μL aliquot of the narrow fibrils sample was applied to each "baked" Quantifoil 1.2/1.3 μm, 200 mesh grid. The grid was then blotted and plunged into liquid nitrogen-cooled liquid ethane in a Vitrobot Mark IV (FEI, Hillsboro, OR, USA) machine[22]. Cryo-EM data were acquired in a Titan Krios microscope (FEI, operated at 300 kV high tension and ×130,000 nominal magnification) equipped with a Quantum LS Imaging System (Gatan, Pleasanton, CA, USA; energy filter slit width was set to 20 eV and K2 camera at counting mode; calibrated pixel size is 1.07 Å). The microscope was aligned as previously described[23]. Data collection was automated by Leginon software package[24]. The defocus value target was set to a single value of 2.7 μm. Dose-fractionation movies were recorded at a frame rate of 5 Hz for a total duration of 10 s. The dosage rate was targeted at 6 electrons (e)/(Å² s), as initially measured by Digital Micrograph (Gatan) software, though fluctuations (within ±10%) in dosage potentially due to electron source instability were subsequently noticed during the imaging session of about 2 days.

Frames in each movie were aligned and summed to generate a micrograph as previously described[25]. Micrographs generated by summing all frames were used to determine defocus values and particle locations by CTFFIND4 (ref.[26]) and manual picking, respectively. We used the micrographs generated by summing the 3rd (5 e/Å²) through the 20th frames (accumulated dose 30 e/Å²) for data processing. Micrographs with severe astigmatism (>9%), obvious drift, or measured underfocus values outside the allowed range (1.5–4 μm) were discarded.

We manually picked filaments indiscriminately in EMAN[27] helixboxer (see statistics in Table 1). The 2D classification revealed two major populations (rod and twister polymorphs) and several other populations. The other minor populations were too poorly defined to be further characterized, and thus omitted in the analysis. Of the classes suitable for analysis, the relative percentages of the rod and twister polymorphs are ~30 and ~70%, respectively. The number is calculated from the accepted classes using Class2D.

We performed 2D and 3D classifications in GPU-accelerated Relion 2.0 (ref.[28]) to separate the particles belonging to the rod and twister polymorphs into subsets as reported previously[25]. We also performed 2D classification of segments extracted with a very large (1024 pixels) box size to determine the pitches of the two polymorphs. Helical parameters were deduced from these pitches with the assumption that each helix had a twisted twofold screw axis. The initial models for the 3D classifications and reconstructions were generated by running Class3D with 1 class, an elongated Gaussian blob as the starting reference, and a fixed helicity based on the above-mentioned assumption. Specifically, the rod class was separated with solely Class2D, and the twister class was separated with Class2D followed by Class3D similar to previously described[25].

We further refined the 3D reconstructions of the rod and twister filaments using Class3D in Relion as previously described[25], except that we now used version 2.0 of Relion with built-in real-space helical reconstruction. Briefly, we started a run of Class3D with one class, with low initial $T$ factor (i.e., "--tau-fudge") and larger (7.5°) angular interval. We gradually increased the $T$ factor and reduced the angular interval with close manual monitoring. We eventually reached a $T$ factor of 256 and an angular interval of 0.975° (healpix order 6) for the final map.

We tested the resolutions of the two resulting maps as previously reported[25]. Two types of Fourier shell correlation (FSC) was calculated: one between the map and atomic model and the other between the 3D reconstructions from two-half datasets (Supplementary Fig. 5). The former, map-model FSC evaluation indicates that the resolutions for the rod and twister maps are 3.7 Å based on the FSC = 0.5 criterion[29]. For the latter, we divided the helical particles by even and odd micrographs (to prevent particles from the same fiber contributing to two different reconstructions), and then calculated a 3D reconstruction from each half dataset using the fully refined center and orientations parameters and performed the FSC calculation. (This FSC is thus not gold standard, as the dataset was not divided in the beginning and was refined in its totality.) Therefore, we used the FSC = 0.5 criterion[28] to evaluate the resolution. We did not apply any density-based or model-based mask for the FSC tests, but a spherical mask of a 170-pixel diameter and 10-pixel apodization, after clipping the density map into 192 × 192 × 192 box. This evaluation indicates that the resolutions for the rod and twister maps are 3.5 and 3.6 Å at FSC = 0.5, respectively.

We built atomic models for the two maps and refined them in central nervous system[30,31] and Phenix *phenix*.real_space_refine[32] with the final Relion refined helical parameters as NCS restraints, as previously described[25]. The statistics are summarized in Table 1.

**Structural analysis and energy calculations**. Based on the two cryo-EM structures of aSyn fibrils, energies for the wild-type and familial mutants were calculated with Rosetta[33]. During the energy evaluation, we omitted the contributions from the statistical terms in the Rosetta scoring function which are derived from monomeric proteins. The total score of each structure was calculated by the sum of the physically meaningful energy components (Lennard–Jones interactions, solvation, hydrogen bonding, and electrostatics). Using either rod or twister structure, the contribution of each mutant was evaluated by the score difference between the mutant and the WT.

**Fiber diffraction**. The procedure followed the protocol described by Rodriguez et al.[8]. To replace the solvent with water, the fibril sample (50 μL) were pelleted by centrifugation at 8000 × *g* for 5 min and washed with deionized $H_2O$ for three times. The fibrils were resuspended in 10 μL of $H_2O$, placed between two capillary glass rods, and allowed to air dry. The next day, the glass rods with fibrils aligned in between were mounted on a brass pin for x-ray diffraction. Each pattern was collected using 1.54 Å x-rays produced by a Rigaku FRE + rotating anode generator equipped with an HTC imaging plate at a distance of 150 mm for 5° rotation width. The results were analyzed using the Adxv software[34].

**Cellular toxicity assay**. The protocol was adapted from the Provost and Wallert laboratories[35]. Thiazolyl blue tetrazolium bromide for the 3-(4,5-dimethylthiazol-2-yl)-2,5-diphenyltetrazolium bromide (MTT) cell toxicity assay was purchased from Millipore Sigma (M2128-1G; Burlington, MA, USA). PC12 cells were plated in 96-well plates at 10,000 cells per well in Dulbecco's modification of Eagle's medium (DMEM), 5% fetal bovine serum (FBS), 5% heat-inactivated horse serum, 1% penicillin/streptomycin, and 150 ng/mL nerve growth factor 2.5S (Thermo Fisher Scientific). The cells were incubated for 2 days in an incubator with 5% $CO_2$ at 37 °C. The cells were treated with different concentrations of aSyn fibrils (200, 500,1000, 2000 nM). The aSyn fibrils were sonicated in a water bath sonicator for 10 min before being added to the cells, the same as the fibrils tested in the in vitro seeding experiment. After 18 h of incubation, 20 μL of 5 mg/mL MTT was added to every well and the plate was returned to the incubator for 3.5 h. With the presence of MTT, the experiment was conducted in a laminar flow hood with the lights off and the plate was wrapped in aluminum foil. The media were then removed with an aspirator and the remained formazan crystals in each well were dissolved with 100 μL of 100% DMSO. Absorbance was measured at 570 nm to determine the MTT signal and at 630 nm to determine background. The data were normalized to those from cells treated with 1% sodium dodecyl sulfate (SDS) to obtain a value of 0%, and to those from cells treated with PBS to obtain a value of 100%.

**Fibril seeding experiment in the aSyn biosensor cells**. Based on a published protocol[36], FRET-based aSyn biosensor cells, HEK293T cells expressing disease-associated aSyn A53T mutant fused with CFP or YFP, were grown in DMEM (4mM L-glutamine and 25 mM D-glucose) supplemented with 10% FBS and 1% penicillin/streptomycin. Trypsin-treated HEK293T cells were harvested, seeded on flat 96-well plates at a concentration of $4 \times 10^4$ cells per well in 200 μL culture medium per well, and incubated in 5% $CO_2$ at 37 °C.

After 18 h, aSyn fibrils were prepared by diluting with Opti-MEM™ (Life Technologies, Carlsbad CA, USA) and sonicating in a water bath sonicator for 10 min. The fibril samples were then mixed with Lipofectamine™ 2000 (Thermo Fisher Scientific) and incubated for 15 min and then added to the cells. The actual volume of Lipofectamine™ 2000 was calculated based on the dose of 1 μL per well. After 48 h of transfection, the cells were trypsinized, transferred to a 96-well round-bottom plate, and resuspended in 200 μL

chilled flow cytometry buffer (Hank's balanced salt solution, 1% FBS, and 1 mM EDTA) containing 2% paraformaldehyde. The plate was sealed with parafilm and stored at 4 °C for flow cytometry.

No apparent toxicity is observed at the tested concentrations of aSyn fibrils used in the seeding assay (Supplementary Fig. 13), which rules out the contribution of cell death to aSyn seeding.

**Flow cytometry-based FRET analysis**. Intracellular aSyn aggregation or inclusions were quantified by the flow cytometry-based FRET analysis. The protocol was adapted from the Diamond laboratory[37]. The fluorescence signals of the cells were measured using the settings for CFP (ex. 405 nm, em. 405/50 nm filter), YFP (ex. 488 nm, em. 525/50 nm filter), and FRET (ex. 405 nm, em. 525/50 nm filter) with an LSRII Analytic Flow Cytometer (BD Biosciences). FRET signals were used to differentiate the aggregated aSyn from the non-aggregated aSyn. A bivariate plot of FRET vs. CFP was created to introduce a polygon gate to exclude all of the FRET-negative cells treated with only Lipofectamine and to include the FRET-positive cells treated with fibril seeds (Supplementary Fig. 3). The integrated FRET density, calculated by multiplying the percentage of FRET-positive cells by the mean fluorescence intensity of the FRET-positive cells, was reported in the results.

**Statistical analysis**. All statistical analyses were performed in SigmaPlot version 13.0 (Systat Software Inc., San Jose, CA, USA). The Grubbs' test was used to exclude outliers. One-way analysis of variances were used to assess differences between the fibril-treated and control-treated cells in the in vitro cytotoxicity assay. $P$ values <0.01 were considered statistically significant.

## Data availability

The cryo-EM density maps of the rod and twister polymorphs have been deposited in the Electron Microscopy Data Bank under accession number EMD-7618 and EMD-7619, respectively, with associated atomic coordinates deposited in the RCSB Protein Data Bank under accession number 6CU7 and 6CU8, respectively. Other data are available from the corresponding authors upon reasonable request.

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

## Acknowledgements

We thank Ivo Atanasov for technical assistance in cryo-EM. We acknowledge the use of instruments at the Electron Imaging Center for Nanomachines supported by UCLA and by instrumentation grants from NIH (1S10RR23057 and 1U24GM116792) and NSF (DBI-1338135 and DMR-1548924). We acknowledge computing resource support from XSEDE (MCB140140), which is supported by NSF (ACI-1053575). We also thank the Diamond Laboratory (UT Southwestern) for providing the aSyn biosensor cells. This work was supported by departmental recruitment funds to L.J. This research was also supported in part by National Institutes of Health (AG029430 to D.S. E., GM071940 and AI094386 to Z.H.Z.). K.A.M. is supported by the UCLA Medical Scientist Training Program (GM08042) and UCLA Chemistry-Biology Interface training grant (USPHS National Research Service Award 5T32GM008496). D.S.E. is supported by the Howard Hughes Medical Institute (HHMI).

## Author contributions

L.J., Z.H.Z., and D.S.E. designed and supervised the research. B.L. characterized the aSyn fibrils, performed in vitro seeding and toxicity experiments, and analyzed data. P.G. performed the cryo-EM studies, processed data, and calculated the maps. K.A.M. and M.R.S. refined the models and simulated fiber powder diffraction. D.R.B. assisted in cryo-EM data processing. P.G., S.Y., K.A.M., and M.R.S. built atomic models. P.S., B.L., and M.Z. prepared aSyn and optimized fibril growing conditions.

G.N. and W.S.S. helped with in vitro seeding and toxicity experiments. B.L., K.A.M., P.G., D.S.E., Z.H.Z., and L.J. wrote the manuscript with the input from all authors.

## Additional information

**Competing interests:** D.S.E. is an advisor and equity shareholder in ADRx Inc. The other authors declare no competing interests.

