## [Peer Review File · Nature Communications]

Reviewers' comments:

Reviewer #1 (Remarks to the Author):

The manuscript "Cryo-EM of full-length α -synuclein reveals fibril polymorphs with a common structural kernel" by Li et al provides atomic resolution of two polymorphs of α -synuclein fibrils, a rod and twister type. They test the effect of this mix of fibrils on cellular α -synuclein aggregation and cytotoxicity. The authors then use the structures to predict the effect of familial Parkinson's disease mutations on either structure, and predict that disease-linked mutations can change the composition of polymorphic α -synuclein fibrils, by altering the populations of certain polymorphs, which may have consequences on pathogenicity.

The study is interesting and timely, as only recently it has emerged that different forms of α -synuclein fibrils exist, which have different seeding capability and potentially different toxicity. This may explain differences in disease onset and severity. Also, modeling the effect of Parkinson's disease-linked mutations on these structures is interesting, as it may shed light on their increased pathogenicity. However, the cell biological aspects of the study and the link to fibrils seen in patient brains are weak and require further clarification and experimentation.

The authors should address the following points:

- The authors use recombinant α -synuclein produced in bacteria to produce an ensemble of α -synuclein fibril polymorphs under a variety of conditions. There is no mentioning as to whether the human sequence was used, which conditions give rise to which fibrils, and which of these conditions was used for the rest of the manuscript. This information should be included in Fig. S1.
- The authors proceed with α -synuclein fibrils generated in tetrabutylphosphonium bromide, which give rise to 12 nm-wide fibrils of non-twisted or twisted rod shape. The relevance to fibrils seen in patients is worrisome because (a) the fibril widths detected in patients is 10 nm and 5 nm, respectively, and (b) the conditions of generation of these fibrils is very artificial. A structural comparison of the two fibril types here and fibril structures seen in patients would be useful.
- Fig. 1a: What is the relative percentage of these two fibril polymorphs in the population mix?
- Fig. 1c: The authors should provide a better explanation of the HEK FRET biosensor cells and how this assay was performed. E.g. why is there already massive fluorescence signal and potentially α -synuclein aggregation in controls, what are the criteria for FRET analysis inclusion/exclusion in panel d if any at all, can the resolution of these images be improved so cellular boundaries can be depicted? This is particularly important to analyze potential cytotoxicity at 200 nM fibril concentration where there seem to be less cells. Also, large dots may be rounded up cells or dead cells. A viability assay would be useful (see also comment below).
- Fig. 1c and e: It is unclear why the authors have chosen two different cellular systems to measure aggregation and cell toxicity. In panel c, the authors use full length fibrils and transfect them with lipofectamine. In panel e, the authors use NGF-differentiated PC12 cells and use sonicated fibrils that were added exogenously. It is known that smaller pieces of fibrils are more cytotoxic compared to larger fibrils, so these two assays need to be done under the same condition to draw any conclusions.
- Fig. 1: It is entirely unclear from the cellular assays if both or only one type of α -synuclein fibril polymorph causes aggregation and cytotoxicity.

Minor issues:

- Fig. S3: Please label the right image in panel a with "lipofectamine + fibril seed".

- Fig. 4a: Please show α -synuclein's primary sequence in scale.

Reviewer #2 (Remarks to the Author):

The manuscript from Li et al. reports the 3.7 Å resolution cryo-EM structure of 2 morphology types of alpha-synuclein polymers. Due to the direct involvement of alpha-synuclein in Parkinson's disease the cryo-EM structure is of high general interest. The fibril preparations were optimized in a screen and then tested for cytotoxicity. The peptide chains form a beta-arch core including steric zippers in the two polymorphs. The two polymorphic structures termed rod and twister differ in their protofilament interfaces and the exact number of structured residues that are part of the fibrillar scaffold. The rod polymorph contains a series of disease-related mutations directly at the protofilament interface whereas the twister polymorph does not involve any of these residues in that interface. The manuscript is well written and provides a significant advance in our understanding of disease mutations in the context of polymorphic alpha-synuclein fibrils.

Points

1. During the revision of the paper a competing work appeared in eLife by Guerrero-Ferreira et al., which includes a slightly higher resolution (3.4 Å) cryo-EM structure corresponding to the twister morphology of this work. In the light of the paper, it would be appropriate to add a section on the comparison of the current structures vs. the Guerrero-Ferreira structures in addition to the discussed microED structures.

2. Resolution curve and FSC criterion

There seems to be some issue in the resolution curve the FSC criterion. Resolutions are generally reported at the FSC 0.143 criterion (Rosenthal and Henderson, 2003) whereas the authors chose to report the 0.5 criterion. The presented model map FSC is correctly compared at the 0.5 criterion. The half-map FSCs shown in Fig. S5 are not typical FSC curves and possess some sort of regular oscillation over a large resolution range. The authors should detail the exact procedure how they computed the FSC including masking. Perhaps stringent masks were employed that lead to this unusual FSC curve. Visual judgement of the map from the presented figures, however, confirms the resolution claim between 3.5 and 4.0 Å.

typo:

Methods, page 15: summary of frames sum of frames

Responses to reviewers' comments

Dear Reviewers,

We thank you for the prompt evaluation and valuable comments. As you will see from the itemized responses detailed below, we have addressed the comments thoroughly and revised our manuscript accordingly. In particular, we performed a new experiment of viability assay of HEK biosensor cells, added a comparison with recently published structure of a truncated aSyn in our Discussion, and expanded method sections concerning fibril preparation, HEK biosensor cells, toxicity assay, 3D reconstruction and resolution evaluation.

Reviewers' comments:

Reviewer #1 (Remarks to the Author):

The manuscript “Cryo-EM of full-length α -synuclein reveals fibril polymorphs with a common structural kernel” by Li et al provides atomic resolution of two polymorphs of α -synuclein fibrils, a rod and twister type. They test the effect of this mix of fibrils on cellular α -synuclein aggregation and cytotoxicity. The authors then use the structures to predict the effect of familial Parkinson’s disease mutations on either structure, and predict that disease-linked mutations can change the composition of polymorphic α -synuclein fibrils, by altering the populations of certain polymorphs, which may have consequences on pathogenicity.

The study is interesting and timely, as only recently it has emerged that different forms of α -synuclein fibrils exist, which have different seeding capability and potentially different toxicity. This may explain differences in disease onset and severity. Also, modeling the effect of Parkinson’s disease-linked mutations on these structures is interesting, as it may shed light on their increased pathogenicity.

However, the cell biological aspects of the study and the link to fibrils seen in patient brains are weak and require further clarification and experimentation.

The authors should address the following points:

- The authors use recombinant α -synuclein produced in bacteria to produce an ensemble of α -synuclein fibril polymorphs under a variety of conditions. There is no mentioning as to whether the human sequence was used, which conditions give rise to which fibrils, and which of these conditions was used for the rest of the manuscript. This information should be included in Fig. S1.

Ans: The full-length aSyn sequence in the study is from human, as specified previously in the results section (Lines 64-66). We now emphasize this in the abstract and Supplementary Fig. 1. We expanded the Methods section (Lines 250-254) to describe the fibril growth conditions that we screened. Out of several hundred conditions, one fibril growth condition (300 μ M aSyn, 15mM tetrabutylphosphonium bromide, room temperature) was selected for further characterization. This information is also included in Supplementary Fig. 1.

- The authors proceed with α -synuclein fibrils generated in tetrabutylphosphonium bromide, which give rise to 12 nm-wide fibrils of non-twisted or twisted rod shape. The relevance to fibrils seen in patients is worrisome because (a) the fibril widths detected in patients is 10 nm and 5 nm, respectively, and (b) the conditions of generation of these fibrils is very artificial. A structural comparison of the two fibril types here and fibril structures seen in patients would be useful.

Ans: We thank the reviewer for pointing out the discrepancy of the fibril width measurements. Upon re-evaluation of the measurements of the fibril widths in the negative stain EM images, we realized that the outer boundary of the negative stain dye (http://www.snaggledworks.com/em_for_dummies/negative_stain.html) was erroneously included in the initial measurement. As illustrated in the attached image, this choice of boundary led to an over-estimation of the fibril widths in the initial measurement, giving a distribution centered on 12 nm. After careful unbiased re-measurement of the widths of the fibril population, a distribution closer to 10 nm was observed when the stain boundary was not included. We updated Figure 1 and revised the manuscript to reflect the changes. Furthermore, the fibril width of 10 nm is confirmed by our cryoEM study. The cryoEM structures of the rod and twister polymorphs revealed the accurate fibril widths of 96 Å and 99 Å (Figure 2a), respectively, which is consistent with the 10 nm wide fibril found in patients.

The fibril growth condition used in the study was implemented in order to generate a fibril population that was well separated and suitable for structural characterization. While our fibers were grown *in vitro*, they did show biological activity, including cellular toxicity and seeding. We attempted to do a comparison of our fibril preparation with patient-extracted fibrils. The highest resolution structure relevant to patient-extracted aSyn fibrils is the ssNMR structure from recombinant aSyn seeded by patient extracts (Tuttle et al., NSMB 2016). Although the ssNMR structure is not of patient-extracted fibrils, we have compared the structures in the Discussion (Lines 181-184). When comparing our fibril preparation with the patient-extracted fibrils, we found the same fibril width of ~10nm (Line 178-181). But the published EM images of patient-extracted samples (Spillantini et al., PNAS 1998) are not of high enough resolution to reveal any distinguishing features for a detailed structural comparison.

- Fig. 1a: What is the relative percentage of these two fibril polymorphs in the population mix?

Ans: The rod and twister polymorphs are the two major populations in our fibril preparation. Within the two major populations, the relative percentages are ~30% (Rod) and ~70% (Twist). The calculation is described in the Method section (Lines 291-295).

- Fig. 1c: The authors should provide a better explanation of the HEK FRET biosensor cells and how this assay was performed. E.g. why is there already massive fluorescence signal and potentially α -synuclein aggregation in controls, what are the criteria for FRET analysis inclusion/exclusion in panel d if any at all, can the resolution of these images be improved so cellular boundaries can be depicted? This is particular important to analyze potential cytotoxicity at 200 nM fibril concentration where there seem to be less cells. Also, large dots may be rounded up cells or dead cells. A viability assay would be useful (see also comment below).

Ans: We thank the reviewer for the comments. The detailed explanation of the FRET-based HEK biosensor assay was provided in the Results (Lines 78-81) and Method (Lines 388-397) sections. As shown in the updated Figure 1, soluble, endogenously expressed YFP-fused aSyn protein appears as diffuse background fluorescence using the FITC channel from fluorescence images of controls. This diffuse background is expected for the non-aggregated state of YFP-fused aSyn. The aggregated aSyn is visualized as fluorescent puncta (white arrow in Figure 1c), rather than diffuse background fluorescence. The aSyn aggregates were then quantified by flow cytometry-based FRET analysis. The soluble, non-

aggregated aSyn in control had much weaker FRET signals than the aggregated aSyn, as shown in Figure 1D. The criteria for FRET analysis was described in Supplementary Figure 3.

We updated images in Figure 1c to improve visualization of the cell boundaries. Following the reviewer's suggestion, we performed the viability assay of HEK biosensor cells (Supplementary Fig. 13) and confirmed that there is no apparent cytotoxicity observed at various fibril concentrations. Among the fibril concentrations (up to 200 nM) that we tested in *in vitro* seeding experiments, aSyn fibrils exhibited no significant cytotoxicity, which rules out the contribution of cell death to aSyn seeding.

- Fig. 1c and e: It is unclear why the authors have chosen two different cellular systems to measure aggregation and cell toxicity. In panel c, the authors use full length fibrils and transfect them with lipofectamine. In panel e, the authors use NGF-differentiated PC12 cells and use sonicated fibrils that were added exogenously. It is known that smaller pieces of fibrils are more cytotoxic compared to larger fibrils, so these two assays need to be done under the same condition to draw any conclusions.

Ans: The logic behind the use of two different cellular systems to measure aggregation and cytotoxicity is the following: cell death may potentially contribute to aSyn aggregation or seeding (Sabate, Prion 2014; Busquet et al., Biomed Res Int 2015). Thus, to measure seeding we used the HEK293T cell line (Holmes et al., PNAS 2014) which showed no significant cytotoxicity at a wide range of fibril concentrations tested in the assay. To obtain a more biologically relevant assay for cytotoxicity, we used a neuron-like cell line of NGF-differentiated PC12 to study aSyn toxicity.

We performed the two assays of seeding and toxicity using the same sonicated fibrils, which agrees with the reviewer that “two assays need to be done under the same condition to draw any conclusions”. In the seeding assay, the fibrils of human full-length aSyn (1-140) were sonicated and transfected into the HEK293 cells. In the toxicity assay, the same sonicated fibrils with different concentrations were added into the PC12 cells. For clarification, we revised the Results (Lines 85-86) and Method (Lines 357-358) sections accordingly.

- Fig. 1: It is entirely unclear from the cellular assays if both or only one type of α -synuclein fibril polymorph causes aggregation and cytotoxicity.

Ans: The twister and rod polymorphs coexist in our fibril preparation. We have made many attempts to “isolate” either polymorph. But varying the buffer condition did not yield a pure population of a single polymorph. The seeding and cytotoxicity experiments using the fibril preparation with the mixture of

both polymorphs cannot quantify the individual contributions from either polymorph. However, we are able to tell that at least one of the polymorphs in the preparation is cytotoxic/seeding competent. Using structure analysis of disease-related familial mutations, together with the seeding and toxicity experiments, we are able to make a hypothesis as to which polymorph may be responsible for the toxicity and seeding we observe (Lines 211-221).

Minor issues:

- Fig. S3: Please label the right image in panel a with “lipofectamine + fibril seed”.

Ans: Fixed.

- Fig. 4a: Please show α -synuclein's primary sequence in scale.

Ans: Fixed.

Reviewer #2 (Remarks to the Author):

The manuscript from Li et al. reports the 3.7 Å resolution cryo-EM structure of 2 morphology types of alpha-synuclein polymers. Due to the direct involvement of alpha-synuclein in Parkinson's disease the cryo-EM structure is of high general interest. The fibril preparations were optimized in a screen and then tested for cytotoxicity. The peptide chains form a beta-arch core including steric zippers in the two polymorphs. The two polymorphic structures termed rod and twister differ in their protofilament interfaces and the exact number of structured residues that are part of the fibrillar scaffold. The rod polymorph contains a series of disease-related mutations directly at the protofilament interface whereas the twister polymorph does not involve any of these residues in that interface. The manuscript is well written and provides a significant advance in our understanding of disease mutations in the context of polymorphic alpha-synuclein fibrils.

Ans: Thank you!

Points

1. During the revision of the paper a competing work appeared in eLife by Guerrero-Ferreira et al., which includes a slightly higher resolution (3.4 Å) cryo-EM structure corresponding to the twister morphology of this work. In the light of the paper, it would be appropriate to add a section on the comparison of the current structures vs. the Guerrero-Ferreira structures in addition to the discussed microED structures.

Ans: We have added a passage of the comparison in our revised manuscript (Lines 184-186). The Guerrero-Ferrera et al. paper reports one structure of a truncated aSyn (1-121) fibril. The structure bears topological similarity to our rod polymorph of full-length human aSyn (1-140) fibril. Yet there are appreciable differences between the atomic models, with an RMSD of 2.1 Å for their C α backbone atoms. The second structure reported in our paper, i.e., the twister polymorph, displays much less similarity with significant deviation of packing and backbone from the structure reported in the Guerrero-Ferrera et al. paper.

2. Resolution curve and FSC criterion

There seems to be some issue in the resolution curve the FSC criterion. Resolutions are generally reported at the FSC 0.143 criterion (Rosenthal and Henderson, 2003) whereas the authors chose to report the 0.5 criterion. The presented model map FSC is correctly compared at the 0.5 criterion. The half-map FSCs shown in Fig. S5 are not typical FSC curves and possess some sort of regular oscillation over a large resolution range. The authors should detail the exact procedure how they computed the FSC including masking. Perhaps stringent masks were employed that lead to this unusual FSC curve. Visual judgement of the map from the presented figures, however, confirms the resolution claim between 3.5 and 4.0 Å.

Ans: The reviewer has a good point of the unusual FSC oscillation. We have expanded the method section describing the 3D reconstruction and FSC evaluation substantially (Lines 283-325). In the revised method, we indicated the reason behind our cautionary use of the 0.5 FSC criterion because of structure refinement using the full dataset instead of totally independent refinement using half datasets and cited the original paper documenting such usage (Scheres and Chen, 2012 Nature Methods). As the reviewer rightly pointed out, we relied on the more accurate estimator of the FSC=0.5 criterion between the map and the model, which was calculated and reported in Supplementary Fig. 5 (orange curves). In addition, this estimate is consistent with structural features in the maps.

The oscillation of the FSC over a large resolution range is likely due to the presence of structural repeats in the aSyn fibril structure. For example, the FSC has a prominent rise around 4.8 Å (orange curve, Supplementary Fig. 5) due to the fact that the structure is predominantly β sheets spaced at that distance.

typo:

Methods, page 15: summary of frames sum of frames

Ans: Fixed. Lines 283, 286.

REVIEWERS' COMMENTS:

Reviewer #1 (Remarks to the Author):

In the revised manuscript „Cryo-EM of full-length alpha-synuclein reveals fibril polymorphs with a common structural kernel“, Li et al have addressed most of my concerns.

Remaining issue are:

Fig. 1c-e: It remains unclear to me why the authors have transfected sonicated fibrils into HEK cells to measure seeding and aggregation, while they add sonicated fibrils endogenously to N2a cells to measure cytotoxicity. Transfection is likely to result in much more intracellular fibril concentration and exaggerated toxicity.

Fig. 4a: The arrows highlighting human missense mutations do not align with the provided sequence.

Responses to reviewers' comments

Dear Reviewers,

Please find our itemized responses detailed below. We thank you for the prompt evaluation and valuable comments.

Reviewers' comments:

Reviewer #1 (Remarks to the Author):

In the revised manuscript, Cryo-EM of full-length alpha-synuclein reveals fibril polymorphs with a common structural kernel“, Li et al have addressed most of my concerns.

Ans: Thank you.

Remaining issue are:

Fig. 1c-e: It remains unclear to me why the authors have transfected sonicated fibrils into HEK cells to measure seeding and aggregation, while they add sonicated fibrils endogenously to N2a cells to measure cytotoxicity. Transfection is likely to result in much more intracellular fibril concentration and exaggerated toxicity.

Ans: As we stated in our previous response, cell death may potentially contribute to aSyn aggregation or seeding (Sabate, Prion 2014; Busquet et al., Biomed Res Int 2015). We aimed to perform two separate experiments to assess seeding/aggregation and cytotoxicity, independently.

In the seeding/aggregation experiment, we sought a cellular assay at a range of tested aSyn fibril concentrations without significant toxicity in order to avoid the potential complication from toxicity. Transfection of sonicated fibrils into HEK cells is a well-established method to measure seeding and aggregation (Holmes et al., PNAS 2014). Transfection, which results in more intracellular fibril concentration, requires lower effective concentration of aSyn fibrils in the assay. At this concentration we observed no cytotoxicity. We also tried the seeding assay without transfection and a much higher concentration of aSyn fibrils (~100x more concentrated) is needed to achieve significant seeding. But that high concentration of aSyn fibrils leads to obvious cell death, which could potentially confound the seeding/aggregation observed. We chose to present the seeding experiments with transfection because the effective concentration of aSyn fibrils showed no significant toxicity.

In the cytotoxicity experiment, the addition of sonicated fibrils to the NGF-differentiated PC12 cells is a well-established assay to measure cytotoxicity of amyloid fibrils (Wakabayash & Matsuzaki, *Journal of Molecular Biology* 2007; Ono et al., *PNAS* 2009). Use of this neuron-like cell line allows us to obtain a more biologically relevant assay for cytotoxicity.

Fig. 4a: The arrows highlighting human missense mutations do not align with the provided sequence.

Ans: Fixed.